# Throughput Enhancement in Downlink MU-MIMO Using Multiple Dimensions

**Jong-Gyu Ha, Jae-Hyun Ro and Hyoung-Kyu Song** *

Department of Information and Communication Engineering, uT Communication Research Institute, Sejong University, Gunja-dong 98, Gwangjin-gu, Seoul 05006, Korea
* Correspondence: songhk@sejong.ac.kr; Tel.: +82-2-3408-3890

**Abstract:** This paper focuses on the throughput performance enhancement in the single cell multi-user MIMO (MU-MIMO) downlink system model. For better quality of service, this paper proposes the scheme that increases system throughput and improves the spectral efficiency. Specifically, the signal transmission and detection schemes are proposed by using multiple dimensions. At the transmitter side, two dimensions (power and space) are adopted at the same time. To achieve multiple access (MA), the space domain is exploited by using a block diagonalization (BD) precoding technique, and the power domain is exploited to transmit more data symbols. At the receiver, the signal detection structure corresponding to a transmitter is also proposed. In the simulation results, comparisons of throughput performance are presented in various aspects. As a result, the proposed scheme outperforms the conventional schemes using only one dimension in terms of throughput. This paper shows strong performance in MU-MIMO senarios by adopting multiple dimensions.

**Keywords:** multi-user MIMO; space division multiple access (SDMA); block diagonalization (BD); non-orthogonal multiple access (NOMA); broadcast channel

## 1. Introduction

In the future, in order to handle explosive data traffic, studies will probably aim to increase channel capacity and data rate in the overall wireless communication system [1]. Multiple-input multiple-output (MIMO) has been studied in wireless systems since it has dramatic gains in channel capacity [2]. Also, multiuser MIMO (MU-MIMO) has been studied widely as a potential for improving the overall throughput [3–5]. In downlink broadcasting (BC) channel, MU-MIMO is accomplished by multiuser beamforming that eliminates the multiuser interference (MUI) completely [6–9]. A number of users can be served by one base station (BS) simultaneously and the spectral efficiency can be increased. Therefore, the use of space-division multiple access (SMDA) in the downlink channel provides a considerable gain in system capacity. The sum rate of the MU-MIMO broadcast channel is achieved by dirty paper coding (DPC). However, the critical drawback of DPC is extreme high complexity to implement in practice [10]. Another promising technique in an MU-MIMO system is block diagonalization (BD). In this paper, the proposed scheme considers BD as a generalization of the channel inversion [11,12]. BD supports the multiple data stream with low complexity and approaches the sum capacity of DPC using user selection algorithms [13].

On the other hand, non-orthogonal multiple access (NOMA) is one of the most promising techniques for improving the overall spectral efficiency [14,15]. NOMA shares the same resources with multiple users by exploiting the domains. The well-known NOMA schemes can be divided into power-domain and code-domain NOMA. Power-domain NOMA multiplexing is achieved by different allocated power for users according to the channel conditions. Symbols are superposed to each user and receivers perform successive interference cancellation (SIC) [16]. Furthermore,

MIMO-NOMA is another popular technique to increase sum capacity. In MIMO-NOMA work, the signal processing techniques are investigated and two main routes exist: single-cluster and multi-cluster MIMO-NOMA [17–19]. Using MIMO-NOMA, a full benefit of system capacity and user fairness is achieved by well-chosen power allocation, user clustering, and beamforming [20,21]. Most of these papers consider the user grouping and clustering. However, the proposed scheme fundamentally does not consider user grouping and clustering. Due to the fundamental difference, the proposed scheme has additional advantages. In this paper, the proposed signal transmission and detection scheme use superposition coding (SC) and SIC in conjunction with MU-MIMO to improve the system spectral efficiency. With the SDMA scheme, a number of data symbols can be served to each user by SC and SIC. The power and space domain are fully exploited at the same time and the overall system throughput can be improved in single cell MU-MIMO scenario. The main contributions of this paper are summarized as follows:

1. First, the proposed scheme is written in a different view from the existing MU-MIMO papers. Recently, many researches in MU-MIMO and NOMA have been mainly focused on increasing spectral efficiency by using user clustering, power allocation, and beamforming. User clustering means that close and far users are clustered within each beam and result in intra-beam interference. Intra-beam interference is a major cause of performance degradation. In existing papers, the resource reuse approach is highly affected by intra-beam interference and is critical to overall system performance. However, the proposed scheme does not need to consider intra-beam interference. The proposed scheme makes a single user in each beam and power domain is exploited in a single user. In conclusion, power and spatial domains are exploited within a single user.
2. Second, the proposed scheme improves performance through joint design that uses dimensions appropriately according to wireless communication systems. In a wireless communication system, it is important to decide which technique to use according to the system requirements and characteristics along with cost and complexity constraints. While there are various combinations of techniques that make up the wireless communication system, it is difficult to get a clear answer as to the best scheme for a complex multiuser system under a range of typical operating conditions. This paper proposes a technique that can appropriately bring out the merits of various combinations of techniques.
3. Finally, the proposed idea can be described from a different perspective and provides insights into other system implementations. The key idea here is to use multiple dimensions appropriately. The beauty of this approach is that it can even be applied to other multiple access systems. The various system models can be implemented by using additional dimensions. This paper demonstrates that the system throughput can be increased by using multiple dimensions. This concept can be extended further. If the additional dimensions are used without degradation or with a little tradeoff, the system throughput can be increased. The idea presented in this paper provides a multitude of interesting avenues for future research.

The remainder of this paper is organized as follows. Section 2 describes the problems of the existing scheme and explains the solution and the reasonability of the proposed scheme. Section 3 describes the overall system model and the proposed scheme. Also, this section presents the performance of the proposed scheme. Section 4 presents simulation results and compares with the conventional scheme under various conditions. Section 5 considers some extensions about the implementation of the proposed scheme. Finally, conclusion is provided in Section 6.

## 2. Motivation

The novelty of the proposed scheme is to use multiple dimensions as much as possible. Figure 1a,b represent examples of resource usage in NOMA and BD schemes using only one dimension, respectively. Figure 1c represents example of resource usage for the proposed scheme. In Figure 1a,b

which represent existing techniques, the two techniques cannot be simply combined since all existing schemes utilize each technique for multiple access. However, in the proposed scheme, the SC is used to separate each symbol by utilizing the power domain. In conclusion, it is possible to utilize both power and space domain at the same time and improve performance by changing usage in SC application. In addition, through proper combination, the proposed scheme compensates the shortcomings of existing schemes and has advantages. First, overhead of the BS is reduced. A dynamic user scheduling and grouping strategy needs the feedback information at the BS. The proposed scheme can reduce the feedback overhead significantly by not considering intra-beam interference. Second, proper fairness among users is assured. The interference is treated as noise to users and does not guarantee fairness among users. In the proposed scheme, a certain error probability is ensured for all users. Third, the complexity is reduced for each user. In the case of the existing NOMA scheme, it is necessary to decode even if it is not the user's own signal. However, in the proposed scheme, all the decoded symbols become the user's own data. Therefore, it is possible to reduce the complexity of the user in demodulating unnecessary data. As a result, the proposed scheme allows more flexibility in spreading user signals over the multiusers.

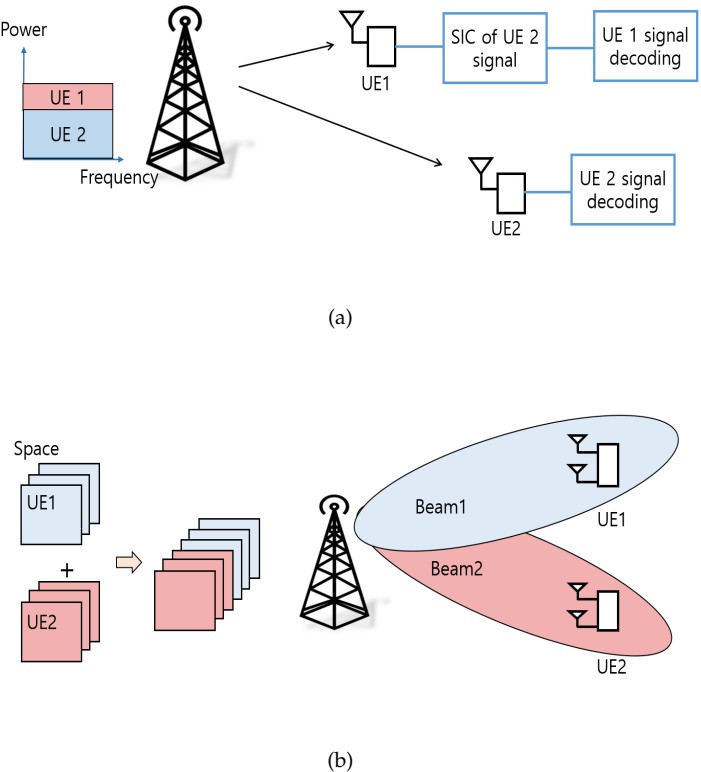

(a)

(b)

**Figure 1.** *Cont.*

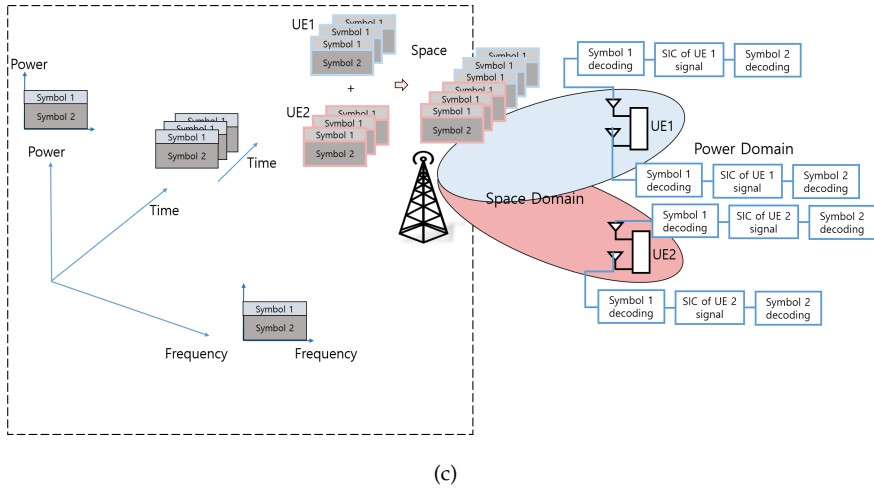

(c)

**Figure 1.** Examples of resource usage in various dimensions (**a**) Example of resource usage for conventional NOMA; (**b**) Example of resource usage for conventional BD; (**c**) Example of resource usage for proposed scheme.

By adopting the multiple dimensions, the same resources can be shared at each domain. In this paper, both the power and space domain are exploited. Figure 2 shows an example of additional dimension usage. Four dimensions (frequency, time, power, space) are used and the same resources can be appropriately shared. Therefore, the more data symbols can be transmitted to more users.

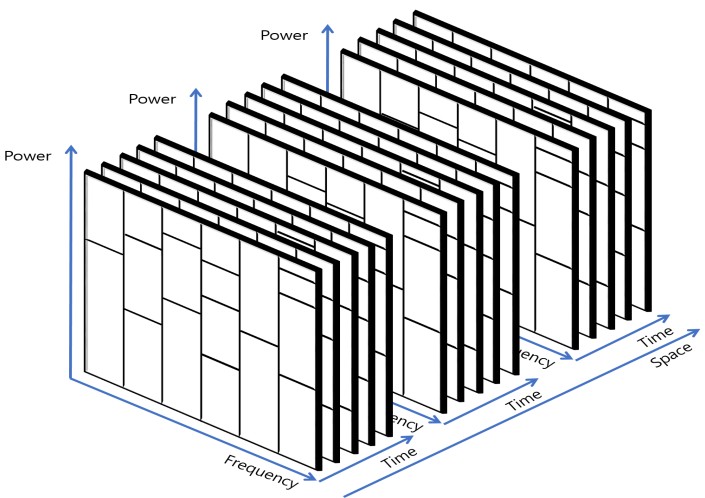

**Figure 2.** The spectral efficiency using multiple dimensions.

This paper considers a downlink single-cell MU-MIMO system as shown in Figure 3. In the proposed scheme, multiple access is accomplished by SDMA and the power domain is exploited to transmit more data symbols to each user. As a result, the proposed scheme has significant potential to improve spectral efficiency and provide better wireless services to many users. Also, this paper offers advantages in various design issues to meet the requirements and characteristics of the system by using multiple dimensions.

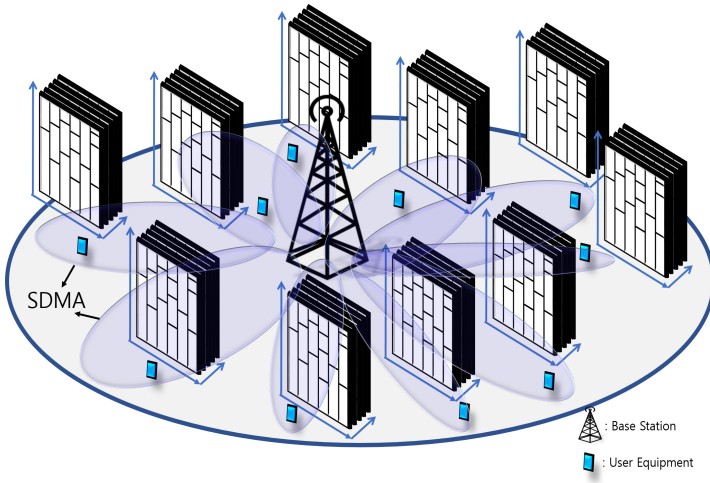

**Figure 3.** The downlink single-cell MU-MIMO system.

## 3. Proposed Scheme

This section describes the proposed scheme from data transmission to signal detection. Using the proposed system, more data symbols can be transmitted and more users can be served simultaneously. First, the transmission model explains how two additional domains can be used to improve spectral efficiency. Second, the signal detection model explains how each signal is detected and reliability can be satisfied. Additionally, in the received SINR, some considerations for the proposed scheme are presented. Finally, the sum throughput between the proposed scheme and conventional schemes is compared. As a result, the throughput performance of the proposed scheme is superior to that of the conventional scheme.

### 3.1. System Model

This paper considers a downlink MU-MIMO broadcast system which consists of one BS and $K$ users as shown in Figure 4. BS is equipped with $N_t$ transmitting antennas and each user has $N_r$ receiving antennas. The MIMO channel of each user is assumed to be flat fading, since frequency selective fading channel can be easily overcome by using orthogonal frequency division multiplexing (OFDM) modulation. The system model can be further extended to frequency selective fading MIMO channel considering all subcarriers. In this system model, the transmit signal for the $k$-th user can be denoted as follows,

$$\mathbf{x}_k = \mathbf{W}_k \mathbf{s}_k. \tag{1}$$

The received signal at the $k$-th user is given by

$$\mathbf{y}_k = \underbrace{\mathbf{H}_k \mathbf{W}_k \mathbf{s}_k}_{\text{desired signal}} + \underbrace{\sum_{j=1, j \neq k}^{K} \mathbf{H}_k \mathbf{W}_j \mathbf{s}_j}_{\text{undesired signals}} + \mathbf{n}_k, k = 1, \cdots, K, \tag{2}$$

where $k$ and $j$ are user indices, $\mathbf{W}_k$ is $N_t \times N_r$ precoding matrix for user $k$, $\mathbf{s}_k$ is a $N_r \times 1$ data symbol vector, $\mathbf{x}_k$ is a $N_t \times 1$ precoded signal vector for the $k$-th user. $\mathbf{y}_k$ is a received signal vector for the $k$-th user and $\mathbf{n}_k$ is $N_r \times 1$ zero-mean additive white Gaussian noise (AWGN) vector with variance $\sigma^2$. In the Equation (2), the first term denotes signal in the intended direction (desired user $\mathbf{s}_k$) and the second term denotes multi-user interference caused due to undesired signals (undesired users $\mathbf{s}_j$)

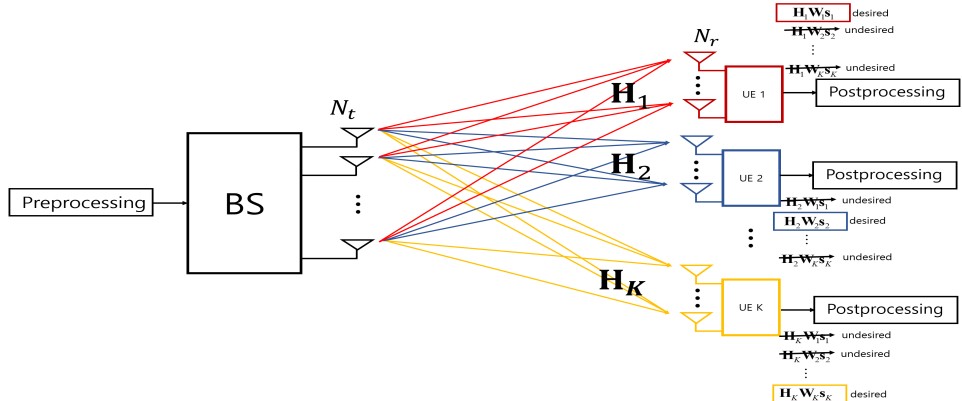

**Figure 4.** The downlink MU-MIMO broadcasting model.

### 3.2. Data Transmission Model

To exploit both the power and space domain, BD and SC scheme are used. The overall transmission model is shown in Figure 5. First, SC is used for exploiting power domain. The transmit symbols are superposed with different powers in one signal. Therefore, the transmitter can transmit more data stream at the same time. In the existing NOMA scheme, SC and SIC are used to suppress the MUI by allocating the different powers to different users. However, the proposed scheme uses SC for separating the symbols in one superposed signal on the same user which BS transmits. In the proposed scheme, the transmit signals for the *i*-th receiving antenna can be written as follows,

$$s_i = \sqrt{P_1}\tilde{s}_{i,1} + \sqrt{P_2}\tilde{s}_{i,2} + \cdots + \sqrt{P_N}\tilde{s}_{i,N}, \tag{3}$$

where *i* is a receiving antenna index, *N* is the number of symbols in one superposed signal. $\tilde{s}$ are the symbols in one superposed signal. The proposed scheme allocates optional power to symbols to detect each symbol, leading $P_1 < P_2 < \cdots < P_N$. Then, transmit signals for the *k*-th user are defined as follows,

$$\mathbf{s}_k = \begin{bmatrix} s_1\,(k) \\ s_2\,(k) \\ \vdots \\ s_{N_r}\,(k) \end{bmatrix} = \begin{bmatrix} \sqrt{P_1}\tilde{s}_{1,1}\,(k) + \sqrt{P_2}\tilde{s}_{1,2}\,(k) + \cdots + \sqrt{P_N}\tilde{s}_{1,N}\,(k) \\ \sqrt{P_1}\tilde{s}_{2,1}\,(k) + \sqrt{P_2}\tilde{s}_{2,2}\,(k) + \cdots + \sqrt{P_N}\tilde{s}_{2,N}\,(k) \\ \vdots \\ \sqrt{P_1}\tilde{s}_{N_r,1}\,(k) + \sqrt{P_2}\tilde{s}_{N_r,2}\,(k) + \cdots + \sqrt{P_N}\tilde{s}_{N_r,N}\,(k) \end{bmatrix}. \tag{4}$$

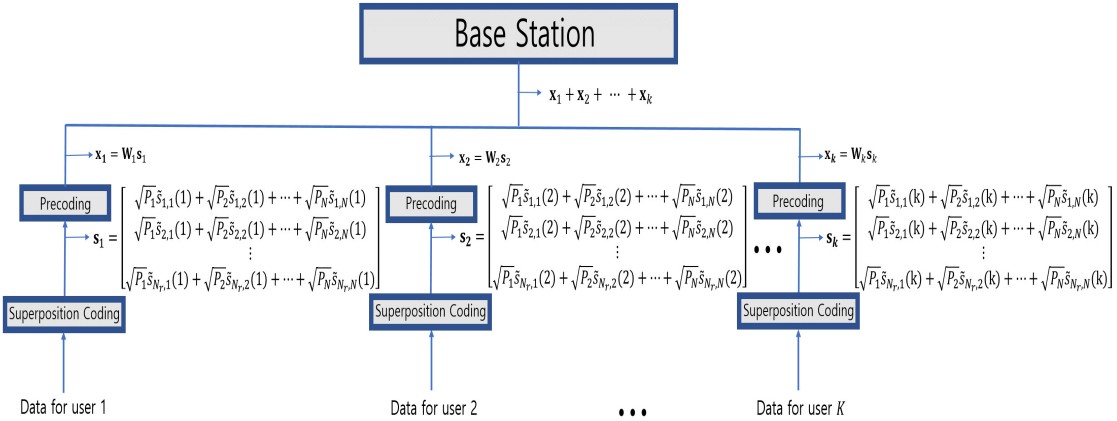

**Figure 5.** The proposed signal transmission model.

For exploiting space domain, the precoding matrix **W** should be designed in SDMA. In the scheme, BD beamforming is adopted to suppress the MUI since BD method shows a strong performance in terms of capacity and has a good flexibility. The objective of the BD method is to completely elminate the MUI by employing the precoding matrix **W**. Then, the precoding matrix design for MUI elimination can be defined as follows,

$$\tilde{\mathbf{H}}_k \mathbf{W}_k = 0, k = 1, \cdots, K. \tag{5}$$

where $\tilde{\mathbf{H}}_k$ is represented as the channel matrix for all users except for user $k$,

$$\tilde{\mathbf{H}}_k = [\mathbf{H}_1^T \cdots \mathbf{H}_{k-1}^T \ \mathbf{H}_{k+1}^T \cdots \mathbf{H}_K^T]^T. \tag{6}$$

With the help of singular value decomposition (SVD), the precoding matrix for eliminating the MUI is designed. SVD is used to decompose a matrix into matrices representing rotation and scaling. By applying the SVD, the $\tilde{\mathbf{H}}_k$ is defined as follows

$$\tilde{\mathbf{H}}_k = \mathbf{U}_k \mathbf{\Lambda}_k [\mathbf{V}_k^{(1)} \ \mathbf{V}_k^{(0)}]^H, \tag{7}$$

where $\mathbf{\Lambda}_k$ is the diagonal matrix of which the diagonal elements are singular value of $\tilde{\mathbf{H}}_k$. $\mathbf{V}_k^{(0)}$ contains vectors of the zero singular values, $\mathbf{V}_k^{(1)}$ and contains vectors of the non-zero singular values. $\mathbf{V}_k^{(0)}$ is an orthogonal basis for the null space of $\tilde{\mathbf{H}}_k$ and the required precoding matrix. As a result, intended user's channel is projected on the null space in order to have the transmission under the constraint of zero-interference (i.e., $\tilde{\mathbf{H}}_k \tilde{\mathbf{V}}_k^{(0)} = 0$, $k = 1, \cdots, K$.)

The received signal of the $k$-th user after eliminating MUI is defined as follows,

$$
\begin{aligned}
\mathbf{y}_k &= \mathbf{H}_k \mathbf{W}_k \mathbf{s}_k + \mathbf{n}_k = \mathbf{H}_{eff,k} \mathbf{s}_k + \mathbf{n}_k \\
&= \mathbf{H}_{eff,k} \begin{bmatrix} s_1(k) \\ s_2(k) \\ \vdots \\ s_{N_r}(k) \end{bmatrix} + \begin{bmatrix} n_1(k) \\ n_2(k) \\ \vdots \\ n_{N_r}(k) \end{bmatrix} \\
&= \mathbf{H}_{eff,k} \begin{bmatrix} \sqrt{P_1}\tilde{s}_{1,1}(k) + \sqrt{P_2}\tilde{s}_{1,2}(k) + \cdots + \sqrt{P_N}\tilde{s}_{1,N}(k) \\ \sqrt{P_1}\tilde{s}_{2,1}(k) + \sqrt{P_2}\tilde{s}_{2,2}(k) + \cdots + \sqrt{P_N}\tilde{s}_{2,N}(k) \\ \vdots \\ \sqrt{P_1}\tilde{s}_{N_r,1}(k) + \sqrt{P_2}\tilde{s}_{N_r,2}(k) + \cdots + \sqrt{P_N}\tilde{s}_{N_r,N}(k) \end{bmatrix} + \begin{bmatrix} n_1(k) \\ n_2(k) \\ \vdots \\ n_{N_r}(k) \end{bmatrix}.
\end{aligned} \tag{8}
$$

where $\mathbf{H}_{eff,k}$ denotes the effective channel of the $k$-th user. According to the Equation (8), the MUI is perfectly eliminated and the $k$-th user receives its own data. Finally, the users can be considered as point-to-point MIMO.

As a result, by exploiting both the spatial and power domains additionally, more data symbols can be transmitted in the same resource (frequency/time). In the proposed scheme, as the number of symbols at each superposed signal is increased, sum throughput at each user is linearly improved. Although the allocated power to each symbol is reduced, the total throughput is improved since the number of transmit symbols is increased. Unlike other techniques that use only one dimension, such as space or power domains, the proposed scheme can achieve significant gains in overall system throughput by using additional domains.

### 3.3. Signal Detection Model

The MUI is perfectly eliminated by using the precoding matrix. Since the $k$-th user receives its own data without MUI, the appropriate receiver structure for each user is similar to the point-to-point MIMO. In Equation (8), MIMO detection is performed. In MIMO detection algorithm, there exist linear and non-linear algorithms. In the proposed scheme, linear detection algorithms such as zero-forcing

(ZF) and minimum mean squared error (MMSE) can be applied simply. If the ZF detection scheme is used, the filter matrix for the *k*-th user is as follows,

$$\mathbf{G}_k = \mathbf{H}_{eff,k}^H (\mathbf{H}_{eff,k}\mathbf{H}_{eff,k}^H)^{-1}. \tag{9}$$

The detected superposed signals of the *k*-th user using ZF MIMO detection can be represented as follows,

$$\hat{\mathbf{s}}_k = \mathbf{G}_k\mathbf{y}_k = \mathbf{H}_{eff,k}^H (\mathbf{H}_{eff,k}\mathbf{H}_{eff,k}^H)^{-1}(\mathbf{H}_{eff,k}\mathbf{s}_k + \mathbf{n}_k)$$
$$= \mathbf{H}_{eff,k}^H \left(\mathbf{H}_{eff,k}^H\right)^{-1} \left(\mathbf{H}_{eff,k}\right)^{-1} \mathbf{H}_{eff,k}\mathbf{s}_k + \mathbf{H}_{eff,k}^H (\mathbf{H}_{eff,k}\mathbf{H}_{eff,k}^H)^{-1}\mathbf{n}_k = \mathbf{s}_k + \mathbf{G}_k\mathbf{n}_k. \tag{10}$$

In the Equation (10), $\mathbf{s}_k$ can be detected since it satisfies $\mathbf{G}_k\mathbf{H}_{eff,k} = \mathbf{I}$. However, if the linear detection algorithms are used, the bit error rate (BER) performance is too poor since the power of symbols in one superposed signal is low. Low performance detection techniques in terms of BER can cause a negative effect on performing SIC. Therefore, the non-linear detection algorithms can be applied such as maximum likelihood (ML), ordered successive interference cancellation (OSIC), decision feedback equalizer (DFE), QRD-M.

From Equation (10), the estimated superposed signals of the *k*-th user after performing MIMO detection can be reconstructed as follows,

$$\hat{\mathbf{s}}_k = \begin{bmatrix} \hat{s}_1(k) \\ \hat{s}_2(k) \\ \vdots \\ \hat{s}_{N_r}(k) \end{bmatrix} = \begin{bmatrix} \sqrt{P_1}\hat{s}_{1,1}(k) + \sqrt{P_2}\hat{s}_{1,2}(k) + \cdots + \sqrt{P_N}\hat{s}_{1,N}(k) \\ \sqrt{P_1}\hat{s}_{2,1}(k) + \sqrt{P_2}\hat{s}_{2,2}(k) + \cdots + \sqrt{P_N}\hat{s}_{2,N}(k) \\ \vdots \\ \sqrt{P_1}\hat{s}_{N_r,1}(k) + \sqrt{P_2}\hat{s}_{N_r,2}(k) + \cdots + \sqrt{P_N}\hat{s}_{N_r,N}(k) \end{bmatrix}, \tag{11}$$

where $\hat{s}$ are the symbols at each estimated superposed signal. In Equation (11), the superposed signal can be decoded by conducting SIC. Therefore, SIC is performed at each receiving antenna. In estimated superposed signal, the strong symbol, i.e., $\sqrt{p_N}\hat{s}_{Nr,N}$ is first decoded. The first decoded symbol can be represented as follows,

$$\hat{\mathbf{s}}_k^d = \begin{bmatrix} \hat{s}_{1,N}^d(k) \\ \hat{s}_{2,N}^d(k) \\ \vdots \\ \hat{s}_{N_r,N}^d(k) \end{bmatrix}. \tag{12}$$

The decoded symbol is then subtracted from the superposed signal.

$$\hat{\mathbf{s}}_k - \hat{\mathbf{s}}_k^d = \begin{bmatrix} \sqrt{P_1}\hat{s}_{1,1}(k) + \sqrt{P_2}\hat{s}_{1,2}(k) + \cdots + \sqrt{P_N}\hat{s}_{1,N}(k) \\ \sqrt{P_2}\hat{s}_{2,1}(k) + \sqrt{P_2}\hat{s}_{2,2}(k) + \cdots + \sqrt{P_N}\hat{s}_{2,N}(k) \\ \vdots \\ \sqrt{P_1}\hat{s}_{N_r,1}(k) + \sqrt{P_2}\hat{s}_{N_r,2}(k) + \cdots + \sqrt{P_N}\hat{s}_{N_r,N}(k) \end{bmatrix} - \begin{bmatrix} \hat{s}_{1,N}^d(k) \\ \hat{s}_{2,N}^d(k) \\ \vdots \\ \hat{s}_{N_r,N}^d(k) \end{bmatrix} = \begin{bmatrix} \hat{s}_{1,N-1}^d(k) \\ \hat{s}_{2,N-1}^d(k) \\ \vdots \\ \hat{s}_{N_r,N-1}^d(k) \end{bmatrix}. \tag{13}$$

Finally, all the symbols in the superposed signal can be decoded by performing SIC. The receiver model is described in Figure 6.

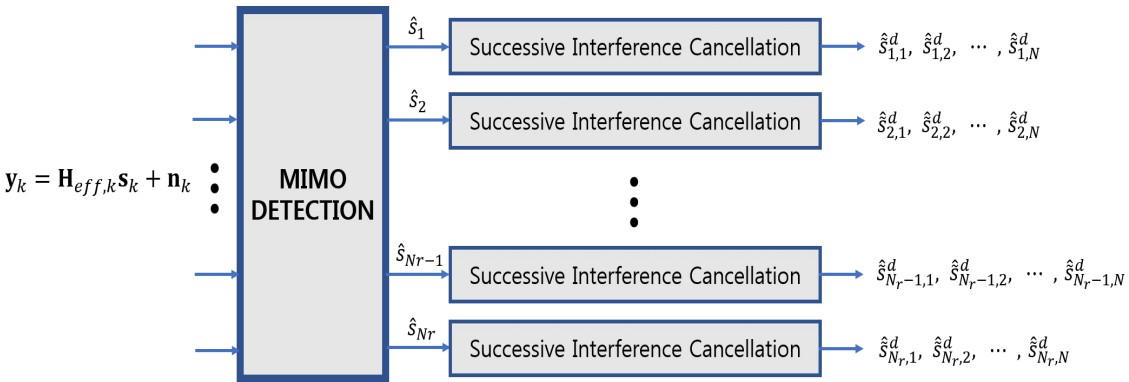

**Figure 6.** The proposed signal detection model.

### 3.4. Received SINR

In this subsection, signal to interference plus noise ratio (SINR) for the symbols in a superposed signal is represented to consider the performance of proposed scheme. If it is assumed that the two symbols are superposed in one superposed signal, i.e., $s_i = \sqrt{P_1}\tilde{s}_{i,1} + \sqrt{P_2}\tilde{s}_{i,1}$, the received signal at the *i*-th receiving antenna for each user after eliminating MUI can be defined as follows,

$$y_i = \lambda_i(\sqrt{P_1}\tilde{s}_{i,1} + \sqrt{P_2}\tilde{s}_{i,2}), \tag{14}$$

where $i$ is received antenna index, $\lambda_i$ is a channel gain in one MIMO parallel channel. In this case, $P_1 < P_2$ subject to total power constraint. Therefore, $\tilde{s}_{i,1}$ is a weak symbol and $\tilde{s}_{i,2}$ is a strong symbol. Then the received SINR for the strong symbol is defined as follows,

$$\text{SINR}_{\tilde{s}_2} = \frac{\lambda_i P_2}{\sigma^2 + \lambda_i P_1}. \tag{15}$$

The weak symbol can be decoded by employing SIC. If it assumed that perfect SIC decoding is conducted, the received SINR for weak symbol is defined as follows,

$$\text{SINR}_{\tilde{s}_1} = \frac{\lambda_i P_1}{\sigma^2}. \tag{16}$$

In the proposed scheme, SINR shows two negative effects in terms of system throughput. First, the weak symbol acts as an interference for received SINR of the strong symbol. As shown in Equation (15), $\lambda_i P_2$ term is considered as noise. Second, strong symbols is likely be decoded incorrectly.If a strong symbol is not properly decoded, it can adversely affect the decoding of weak symbol. The negative effects for the system throughput are restated in the simulation result. Although there are some degradations in terms of throughput on exploiting the power domain, the system throughput is increased since the number of transmit symbols is linearly increased at high SNR.

## 4. Simulation Results

This section shows simulation results to demonstrate the throughput gain of the proposed scheme and compares the results with other conventional schemes. Simulation results also provide some considerations for the proposed scheme. The simulations are performed in seven multi-path Rayleigh fading and time-invariant channel model. OFDM overcomes the frequency selectivity of the wideband channel and multiple carriers enable the high rate data transmission [22,23]. The OFDM symbol is composed of 128 FFT size, four pilots and the 108 data subcarriers based on specification for IEEE 802.11n. The remaining 16 subcarriers are zero padding and OFDM symbol duration is 4 microseconds.

All the simulation are simulated with quadrature phase-shift keying (QPSK) modulation (data bits per subcarrier is 2).

Figure 7a,b compare the sum throughput between the proposed schemes and the conventional schemes for the case of two users as a function of SNR. The throughput $T$ is calculated as follows,

$$\begin{aligned}
T &= N_b \times (1 - E)^L \times K \div T_s \\
N_b &= N_s \times O \times N_r \times s, \\
L &= N_s \times O,
\end{aligned}$$
(17)

where $N_b$ is the number of transmit data bits and $L$ is the number of data bits in one OFDM symbol. $E$ is the BER for each user and $N_s$ is the number of data subcarriers. $O$ is the number of data bits per subcarrier and $s$ is the number of symbols in one superposed signal. $T_s$ is the OFDM symbol duration. In the simulation, proposed scheme and conventional BD are the MIMO system that BS and each user have multiple antennas. On the other hand, the conventional NOMA has one antenna at the BS and each user since multiple access is accomplished by the power domain. The proposed scheme and BD scheme are simulated in the case of $N_t = 4$, $N_r = 2$, $K = 2$ $(4, 2, 2)$ and NOMA scheme is $N_t = 1$, $N_r = 1$ $K = 2$ $(1, 1, 2)$. And in the proposed scheme, the number of symbols at each superposed signal is two $(N = 2)$. Therefore, the total number of transmit symbols to each user is 8. The power of each symbol is allocated at a ratio of 8:2 from the total power $P_t = 1$. In MIMO system, the ML MIMO detection is applied before conducting SIC. The summary of the simulation parameters is shown in the Table 1.

**Table 1.** Simulation Parameters.

| Scheme | $N_s$ | $O$ | $N_r$ | $s$ | $K$ |
|---|---|---|---|---|---|
| Conventional BD | 108 | 2 | 2 | 1 | 2 |
| Conventional NOMA | 108 | 2 | 1 | 1 | 2 |
| Proposed Scheme | 108 | 2 | 2 | 2 | 2 |

In the simulations, there are two types of simulation results: with and without SIC error. Figure 7a,b show the simulation result with SIC error and Figure 7c,d show the simulation result without SIC error. Both cases of the proposed scheme outperform the conventional schemes in terms of maximum throughput since the proposed scheme exploits both the spatial and power dimensions. The conventional BD shows better performance than the proposed scheme in terms of BER. However, the proposed scheme has higher throughput since the proposed scheme transmits more symbols. In Figure 7b, the BER performance of the proposed scheme is better than that of NOMA since the effect of MIMO detection and SIC decoding is improved at high SNR. Figure 7c,d show the impact of SIC error on the proposed scheme. Without SIC error, throughput and BER performance are better than when there is SIC error. If the strong symbol is wrongly decoded at low SNR, the weak signal is also wrongly decoded. As a result, the error propagation occurs. Therefore, the system needs to be designed to avoid error propagation and reduce SIC error. As the proposed scheme minimizes the impact of SIC error, the performance of the proposed scheme approaches the case where there is no SIC error.

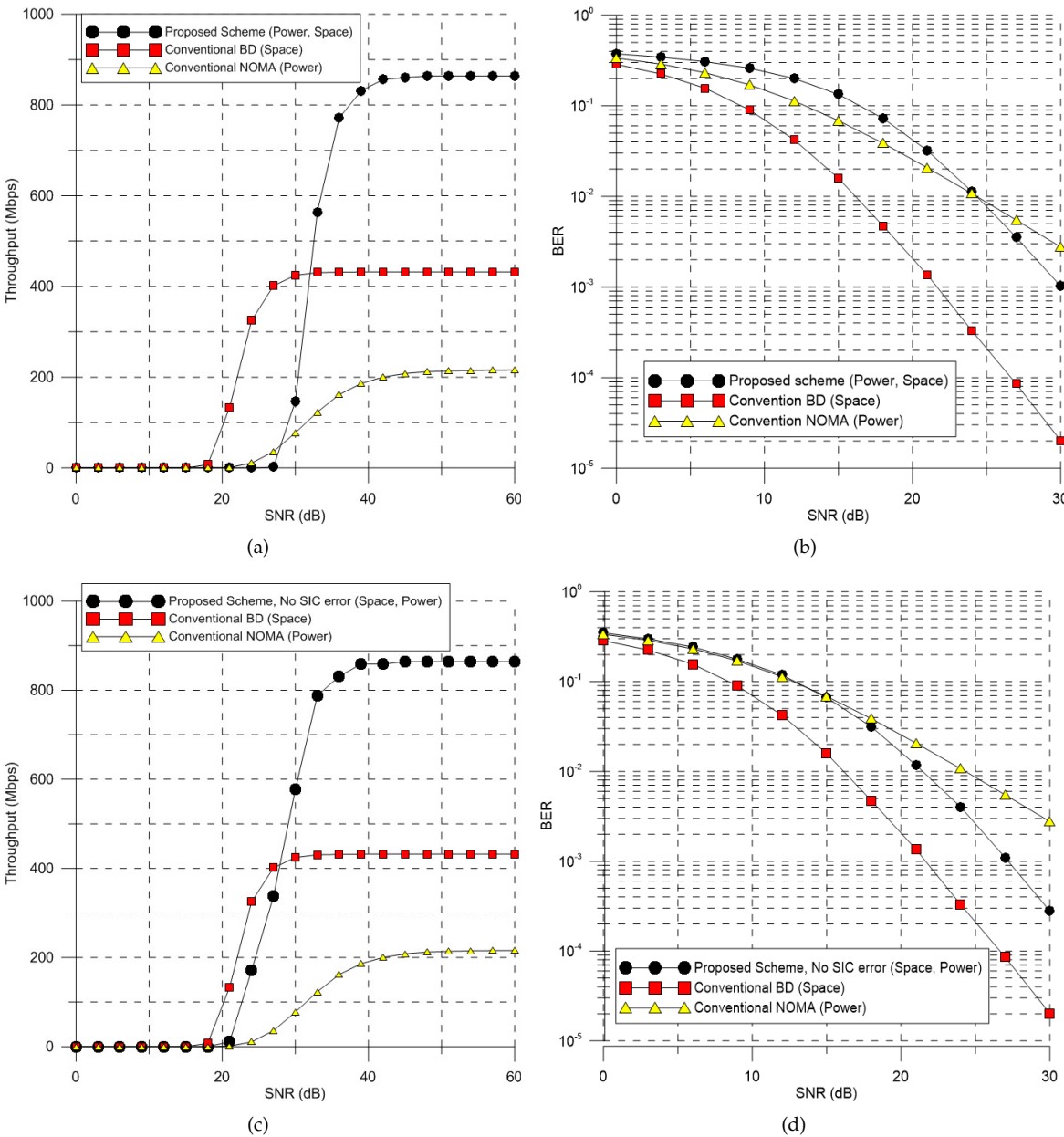

**Figure 7.** Throughput and BER performance of conventional and proposed schemes: (**a**) Throughput performance with SIC error; (**b**) BER performance with SIC error; (**c**) Throughput performance without SIC error; (**d**) BER performance without SIC error.

Figure 8a, b show the performance difference of the proposed scheme according to the number of users and antennas $(N_t, N_r, K)$. The cases of $(6, 3, 2)$ and $(6, 2, 3)$ have the higher throughput performance than $(4, 2, 2)$ case since $(6, 3, 2)$ and $(6, 2, 3)$ transmit more symbols by exploiting both space and power dimension. $(6, 2, 3)$ case has lower throughput performance in low SNR than $(6, 3, 2)$ case since $(6, 3, 2)$ case allocates more symbols to each user. On the other hand, in case of $(6, 2, 3)$, one more user can be serviced. In terms of BER, $(6, 3, 2)$ has better BER performance than $(6, 2, 3)$ since $(6, 3, 2)$ transmits more symbols to each user than $(6, 2, 3)$. $(4, 2, 2)$ has better BER performance than $(6, 2, 3)$ since more power is allocated to each symbol. However, $(6, 2, 3)$ has higher throughput than $(4, 2, 2)$ since $(6, 2, 3)$ transmits more symbols.

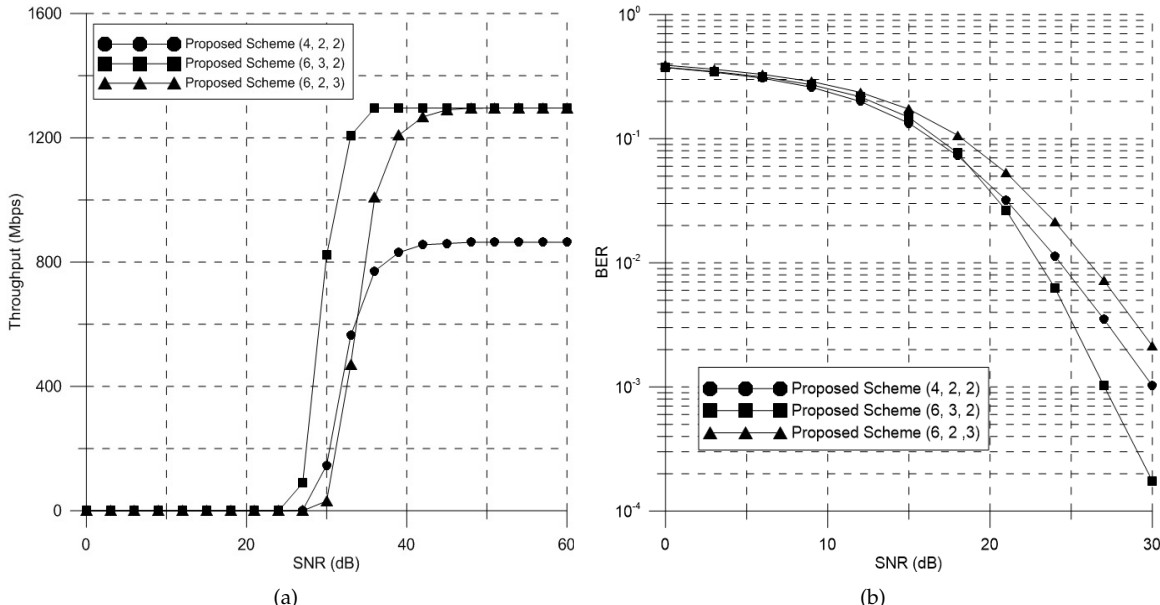

**Figure 8.** Throughput and BER performance of proposed scheme according to the number of users and antennas: (**a**) Throughput performance; (**b**) BER performance.

Figure 9a, b show the difference of the performance between the cases of linear MIMO detection and non-linear MIMO detection. The method with ML detection outperforms the method with ZF detection. The ML technique has better detection performance than the ZF technique before performing SIC on each antenna. Additionally, the ML detection technique mitigates the error propagation compared to the ZF detection scheme.

As a result, the overall simulation results show better performance compared to conventional schemes by using two dimensions simultaneously. In the proposed scheme, by exploiting both power and space domains at the same time, the transmitted symbol is increased. The results show that the superiority of the proposed scheme, and the proposed scheme uses the dimensions appropriately.

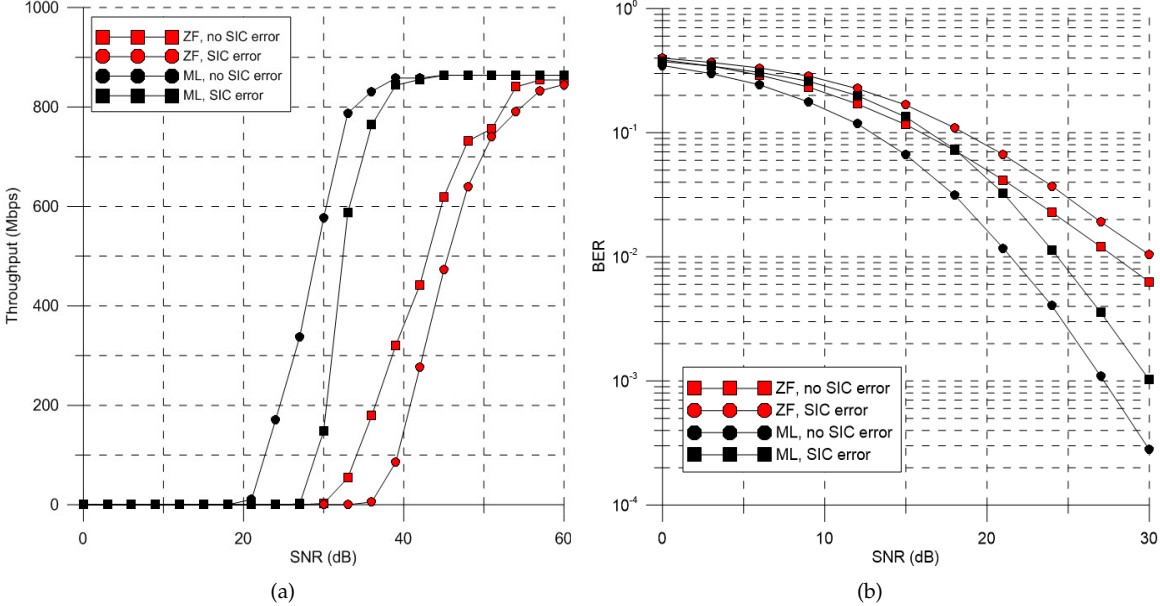

**Figure 9.** Throughput and BER performance of ML detection and ZF detection: (**a**) Throughput performance; (**b**) BER performance.

## 5. Implementation Issue

This section presents some additional considerations for system application and limitation under practical constraints. In addition, this section gives some ideas for additional performance gain and presents some methods to reduce some negative effects in the proposed scheme.

### 5.1. Complexity

In the proposed scheme, the additional implementation complexity in typical NOMA is not needed since SIC is performed for the own user's data. However, even though the superposed signals are the user's own data, the symbols with small allocation power in superposed signal have poor BER performance. To solve this problem, a detection scheme with better performance should be used. However, a non-linear algorithm has high complexity. For this problem, a complexity-reduced detection algorithm can be considered. The main consideration is achieving higher throughput with lower complexity.

### 5.2. SIC-Error Propagation

SIC is often assumed to be successful with perfect decoding. However, for systems with actual modulation and coding, decoding error inevitably occurs, causing error propagation and remarkable performance degradation. As shown in the simulation results, there are performance differences with and without SIC errors. Ultimately, well-designed system for the proposed scheme which avoids error propagation and decoding error should be considered for optimum performance.

### 5.3. Power Allocation

The achievable throughput is affected by the transmit power allocation. If it is assumed that two symbols are superposed in one signal, we should consider how much power should be allocated for each symbol. Basically, allocating more power to strong symbols can reduce the error propagation. However, allocating more power to strong symbols increases the error probability of weak symbols since the power of a weak symbol is too low. As a result, power allocation to each symbol should be considered according to the number of the symbols at the superposed signal subject to total power constraint.

### 5.4. Optimal Parameters

In the proposed scheme, if the number of the symbols in a superposed signal is increased, the sum throughput can be increased linearly. However, the sum throughput cannot be increased without limit since there is a limitation that a receiver can detect the symbols. As more symbols are transmitted, the power allocated to the symbols is reduced. Therefore, if too many symbols are transmitted, the receiver can not detect each symbol. Furthermore, there is also the degradation of sum throughput because of the interference in performing SIC. Therefore, the important issue is to find a near-optimal parameter between the number of users and the number of symbols in a superposed signal in the overall system.

## 6. Conclusions

This paper suggests multi-dimensionality and a methodology to improve the throughput in an MU-MIMO system. This paper also presents the transceiver structure of the proposed scheme. As a result, the multiple dimensions (space and power) are exploited at the same time, and the overasll system spectral efficiency is improved. If more dimensions are used without degradation or with a little tradeoff in performance, the system throughput can be increased. Also, various system models can be implemented by using additional dimensions.

**Author Contributions:** J.-G.H. proposed throughput enchancement scheme for MU-MIMO downlink channel and processed the simulation; J.-H.R. analyzed the simulation results and made the figure; H.-K.S. reviewed

the algorithm and provided the experimental materials for better computational simulations and revised critical errors of the manuscript

**Funding:** This research was supported by Institute for Information & communications Technology Promotion(IITP) grant funded by the Korea government(MSIT) (No.2017-0-00217, Development of Immersive Signage Based on Variable Transparency and Multiple Layers) and was supported by the MSIT(Ministry of Science and ICT), Korea, under the ITRC(Information Technology Research Center) support program(IITP-2018-2018-0-01423) supervised by the IITP(Institute for Information & communications Technology Promotion).

**Conflicts of Interest:** The authors declare no conflict of interest.

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
