# Peer review of "Throughput Enhancement in Downlink MU-MIMO Using Multiple Dimensions"

_electronics, doi:10.3390/electronics8070758_

Round 1
Reviewer 1 Report
The modifications introduced by the authors in the revised version clearly help to understand the novelty of the paper.
Please not use “And “ after a full stop, e.g.,
“The Figure 1 and Figure 2 represent conventional NOMA and BD scheme using only one dimension respectively. And The Figure 3 represents the proposed scheme. “
Reviewer 2 Report
The manuscript has been improved and the concern related to the lack of innovations clarified by the authors. I have no further concerns.
Reviewer 3 Report
The paper considers MU-MIMO together with NOMA to pack additional data transmissions over the radio resources. Essentially, in addition to the spatial domain, the power domain is viewed as a new dimension and multiple data symbols can be overlapped to increase the channel utilization. Authors claim this is an innovation compared to the conventional MU-MIMO scheme and the NOMA scheme, and stated as the main contribution of the paper. However, this is not the case if a comprehensive literature review is carried out. For example, the papers listed at the end of this review have already discuessed the hybrid MU-MIMO/NOMA strategy. To me, this paper simply combines some existing techniques together and do not provide new insight on the harmony among them. At least, I don't see such arguments from the summary of contributions.
Fig 1 and 2 are misleading as the system model hereafter only considers a narrow band flat fading channel, which does not take the time and frequency domains into account.
On the Performance of Beam Division Nonorthogonal Multiple Access for FDD-Based Large-Scale Multi-User MIMO Systems
Yong I. Choi ;
Publication Year: 2017, Page(s): 5077 - 5089
NOMA in Downlink SDMA With Limited Feedback: Performance Analysis and Optimization
IEEE Journal on Selected Areas in Communications
Year: 2017 Volume: 35 , Issue: 10
Pages: 2281 - 2294
Author Response
Please see the attachment.

This manuscript is a resubmission of an earlier submission. The following is a list of the peer review reports and author responses from that submission.
Round 1
Reviewer 1 Report
Reviewer’s Recommendation:
This candidate paper needs minor revisions.
Summary
This work is oriented towards the performance enhancement in a MIMO (MU-MIMO) downlink model involving single cell multi-users. The proposed technique is compared to conventional techniques and is shown as the authors claim, its superiority.
General comments
The paper gives nice ideas relevant to confronting the reduced performance in telecommunication systems by using multidimensionality as the solution. The paper is adequately written while the authors have tried to be simple in their presentation. The hard work is shown. Nevertheless, I propose to the authors to make it even simpler relevant to mathematical relationships as in this way the dissemination would be achievable in additional scientific fields. There are some language mistakes, so I propose this document to be examined by a professional native speaker or nevertheless by the most experienced in English language among the authors team.
Suggested Improvements
Apart from the aforementioned, please apply the following changes and suggestions inside manuscript for the best possible result:
1. Relevant to MIMO and MIMO downlink, please correlate your work if possible to the following works:
- System Performance of an LTE MIMO Downlink in Various Fading Environments In: Volume 70 of the series Lecture Notes of the Institute for Computer Sciences, Social Informatics and Telecommunications Engineering, Chapter 6, http://dx.doi.org/10.1007/978-3-642-23902-1_5
- Design and analysis of a multiple‐output transmitter based on DDS architecture for modern wireless communications. In AIP Conference Proceedings (Vol. 1203, No. 1, pp. 421-426). AIP. http://dx.doi.org/10.1063/1.3322480
2. Relevant to OFDM please do not omit the following:
- OFDM for wireless multimedia communications. Artech House, Inc.
- In depth analysis of noise effects in orthogonal frequency division multiplexing systems, utilising a large number of subcarriers. In AIP Conference Proceedings (Vol. 1203, No. 1, pp. 967-972). AIP. http://dx.doi.org/10.1063/1.3322592
3. Equation (2) should be further explained in simpler manner.
4. You mention that “By applying the singular value decomposition (SVD)”. Please explain.
5. Equation (11): Please elaborate more.
6. Line 101: “λi is a channel gain in one MIMO parallel channel”. Please explain.
7. Lines 109-110: “If the strong symbol is not properly decoded, it can have a negative effect on decoding the weak symbol”. Please explain more about the soft decision method.
8. You mention that “The simulations are performed in 7 multi-path Rayleigh fading and time-invariant channel model”. Is there a special purpose of the utilized model of NLOS? Can be found in bibliography? Please explain.
9. Line 118: “The OFDM symbol is composed of 106 subcarriers”. Please analyze more as we know that IFFT needs in the input powers of two.. Consequently, the OFDM should be consisted of 128 subcarriers with Cyclic Prefix or Zero padding? Please explain.
10. Please revalidate equation (18).
11. Lines 143-145: “The cases of (6, 3, 2) and (6, 2, 3) have the higher throughput performance than (4, 2, 2) case since (6, 3, 2) and (6, 2, 3) transmit more symbols by exploiting both space and power dimension”. Please explain now in simple words what is happening in spatial manner in order to demonstrate the impact of your work.
12. Lines 186-187: “Therefore, if too many symbols are transmitted, the receiver cannot detect each symbol”. Can you explain what the practical limit of such a drawback is?
13. Line 192: “This paper suggests the possibility of multi-dimensionality”. Please revert "This paper suggests the possibility of multi-dimensionality" to "This paper suggests the multi-dimensionality".
14. Please recheck references and convert them strictly to the journal's format.
Manuscript Rating:
The paper should be published after minor corrections.
A future paper should be published involving a potential implementation.
Author Response
The authors uploaded the response to the reviewer's comments as an attachment.

Reviewer 2 Report
The authors combined well known MIMO precoding with SC NOMA scheme. My main concern is the lack of novelty since it is a simple combination of BD precoding with SC NOMA, using decouple well known techniques of both systems. The paper is not well written and organized. Some more concerns:
· Introduction, a paragraph is missing with the paper organization. “This paper is organized as follows….”
· Across the paper you repeat several times that by using several dimensions the spectral efficiency can be improved. This is quite well know, of course using more dimension we have more degrees of freedom to efficiently design the system.
· Section 2 is too small and nothing new is added.
· Fig4. The matrix shown in the figure refers to all users data symbols and not only s1, as is seems that user 1 transmit all symbols in that matrix.
· Section 3.1, The MU-MIMO model is quite well known and therefore I cannot see the interest of Fig. 3
· Why eq(8) is repeated before eq (9)?
· Simulation results. Why have you used a FFT size of 106 (not power of 2)? This value is a little bit strange.
· It would be also interesting present bit error rate results.
Author Response

(The authors gave the same response as above.)

Reviewer 3 Report
The only issue holding this manuscript back from adequately reaching the audience is the very low quality of English usage. A careful effort should be made to improve the Engish writing.
Author Response

(The authors gave the same response as above.)

Round 2
Reviewer 2 Report
The reviewer appreciated the authors’ effort to improve the paper (although the English still poor in some parts). However the fundamental question related with lack of novelty was not answered “My main concern is the lack of novelty since it is a simple combination of BD precoding with SC NOMA, using decouple well known techniques of both systems”. In the reviewer opinion this work is a simple combination of two well know techniques.